# The Association of the Oral Microbiota with the Effects of Acid Stress Induced by an Increase of Brain Lactate in Schizophrenia Patients

**DOI:** 10.3390/biomedicines11020240

**Published:** 2023-01-17

**Authors:** Wirginia Krzyściak, Paulina Karcz, Beata Bystrowska, Marta Szwajca, Amira Bryll, Natalia Śmierciak, Anna Ligęzka, Aleksander Turek, Tamas Kozicz, Anna E. Skalniak, Paweł Jagielski, Tadeusz J. Popiela, Maciej Pilecki

**Affiliations:** 1Department of Medical Diagnostics, Jagiellonian University Medical College, 30-688 Krakow, Poland; 2Department of Electroradiology, Jagiellonian University Medical College, 31-126 Krakow, Poland; 3Department of Toxicology, Jagiellonian University Medical College, Medyczna 9, 30-688 Krakow, Poland; 4Department of Child and Adolescent Psychiatry, Jagiellonian University Medical College, 31-501 Krakow, Poland; 5Department of Radiology, Jagiellonian University Medical College, 31-501 Krakow, Poland; 6Department of Clinical Genomics, Mayo Clinic, Rochester, MN 55905, USA; 7Head and Department of Endocrinology, Jagiellonian University Medical College, 30-688 Krakow, Poland; 8Department of Nutrition and Drug Research, Institute of Public Health, Faculty of Health Sciences, Jagiellonian University Medical College, 31-066 Krakow, Poland

**Keywords:** brain lactate, schizophrenia, oral microbiota

## Abstract

The altered cerebral energy metabolism central to schizophrenia can be linked to lactate accumulation. Lactic acid is produced by gastrointestinal bacteria, among others, and readily crosses the blood–brain barrier, leading to the brain acidity. This study aimed to examine the association of the oral microbiota with the effects of acid stress induced by an increase of brain lactate in schizophrenia patients. The study included patients with a diagnosis of acute polyphasic psychotic disorder meeting criteria for schizophrenia at 3-month follow-up. Results: Individuals with a significantly higher total score on the Positive and Negative Syndrome Scale had statistically significantly lower lactate concentrations compared to those with a lower total score and higher brain lactate. We observed a positive correlation between Actinomyces and lactate levels in the anterior cingulate cap and a negative correlation between bacteria associated with lactate metabolism and some clinical assessment scales. Conclusions: Shifts in the oral microbiota in favour of lactate-utilising bacterial genera may represent a compensatory mechanism in response to increased lactate production in the brain. Assessment of neuronal function mediated by ALA-LAC-dependent NMDA regulatory mechanisms may, thus, support new therapies for schizophrenia, for which acidosis has become a differentiating feature of individuals with schizophrenia endophenotypes.

## 1. Introduction

Schizophrenia is a neuropsychiatric disorder that begins while still in fetal life [1]. From birth through adolescence or early childhood, it can be associated with extensive changes in brain energy metabolism that become central to the pathophysiology of the disease, persisting throughout life [2]. The disease is characterized by a range of symptoms, from positive to negative to cognitive impairment. Positive symptoms include hallucinations, delusions and disorganization of thought and speech. Negative symptoms are associated with apathy, anhedonia, flattened voice and social withdrawal. Emerging cognitive symptoms most commonly relate to impairment of attention, working memory, learning, executive function and social functioning [3]. Understanding the pathophysiology and effectiveness of the therapies used for individual neuropsychiatric disorders is still limited.

Altered brain energy metabolism in schizophrenia can be linked to lactate accumulation and increased acidity, which on the one hand, is explained by the Warburg effect widely described in cancer biology, for which biochemical processes associated with reduced aerobic metabolism, relying on glycolysis as the main energy source, become dominant [4]. Mitochondrial dysfunction—oxidative, nitrosative and sulphur stress associated with insufficient oxygen relative to demand—result in changes in gene expression as key in brain energy metabolism. Consistently, these changes result in reduced brain pH, including changes in neurotransmitter regulation, disruption of mRNA integrity and changes in gene expression patterns. On the other hand, many psychiatric disorders underlain by, among other things, mitochondrial energy dysfunction, increased lactate levels and reduced pH in the brain are explained by increased acidity in the gastrointestinal tract as a result of increased production of lactic acid, a metabolic product of bacteria colonizing the human gastrointestinal tract [2]. The human gut contains more than 1000 unique bacterial species. The relatively less well-studied healthy oral cavity contains approximately 100 to 200 different bacterial organisms [5], of which approximately 25 genera [6] that produce lactic acid are associated with both physical and mental health.

The microbiota of the gastrointestinal tract communicates with the central nervous system through a bidirectional connection between the gut and the brain via the vagus nerve, autonomic nervous system, enteric nervous system, innate and adaptive immunity mechanisms, enteroendocrine signalling, hypothalamic–pituitary–adrenal axis and spinal mechanisms, but also metabolism of neurotransmitters, branched chain amino acids, bile moieties, peptidoglycans and a number of soluble inflammatory mediators [7]. Reciprocal signalling between the gut and brain is regulated at the neural, immunological and endocrine levels [8]. In the case of the oral cavity, this signalling is possible, among others, through branches of the choroid plexus [9]. As a result of this signalling, the effects of the oral microflora on the brain are known to be associated with, among other things, cerebral hypoperfusion, cerebrovascular changes or disruption of the capillary network [10]. Disturbances in the oral microflora can lead to dysbiosis of the gut microflora and immune system disorders [11], which leads to imbalances in central nervous system function and the development of neuropsychiatric, inflammatory, metabolic or autoimmune diseases [12,13,14].

We can consider the oral and intestinal microflora as a whole due to the fact that it forms a feedback loop mediated by the swallowing process, as well as the immune system [15]. The composition of the microflora changes over the course of an individual’s life. The moment of birth is considered to be the beginning, associated with the acquisition and subsequent formation of the child’s microflora based on prenatal and postnatal environmental sensitivities shaping maternal immune system patterns [16,17,18]. The diversity of the microflora depends on many factors, including mode of delivery, past infections, antibiotic therapy, dietary conditions, environmental factors or the genetic susceptibility of the host [19].

The close relationship between the oral and intestinal microflora in schizophrenia and its non-invasive way of assessment justifies the need for more research on the oral versus the intestinal microbiome [20]. Apart from our current study, we found no other original work on the composition of the oral microbiota in patients with schizophrenia and its relationship to clinical, central (brain) and disease-related aspects of the different endophenotypes of the disorder. What is lacking in the work published to date is an assessment of the composition of the oral microbiota versus NMDA-R function in the mechanism of lactate-induced acid stress in schizophrenia. Lactates are a product of metabolism, as well as a substrate utilized by the bacteria of the oral microbiome. Knowledge of what oral bacteria may be commonly found in different groups of patients with schizophrenia, as well as those that may differentiate these patients by central domain, may provide clinically relevant information regarding aetiology, potential diagnostic and prognostic biomarkers, and new therapeutic targets to alter the oral microbiota in a disease, for which negative disorders or cognitive dysfunctions have been shown to be important, in addition to classic positive symptoms.

Lactic acid produced by gastrointestinal bacteria easily crosses the blood–brain barrier [21]. Shifts or interactions between components of the gastrointestinal microbiome may influence the pathophysiology of psychiatric disorders through, among other things, the accumulation of lactate in the brain, which provides support for the so-called glucose alternative-based hypothesis of a neurobiological pathway (related to neuronal activation) using lactate as a primary energy source [22]. Knowledge of the human microbiome may provide an innovative and promising pathway to understanding the pathophysiology of mental illness and potential therapies.

Based on our previous observations, the present study evaluates the association of selected components of the oral microbiota and the products of their metabolism, with central brain changes in patients from specific clinical phenotypes of schizophrenia. In an earlier study conducted by us, changes in glutamatergic transmission divided schizophrenic patients into clinical phenotypes of the illness, which confirmed the diversity of neurochemical changes observed in brain magnetic resonance spectroscopy showing the intrinsic heterogeneity of the disorder [23]. The influence of alterations in glutamatergic transmission is well known from the literature on the subject as being key to the negative or cognitive symptoms of schizophrenia that may appear in the course of the disease, but the molecular mechanism in humans has not yet been elucidated [24].

A small proportion of the glutamate detected by MRS brain magnetic resonance spectroscopy is involved in glutamate neurotransmission [25]. This is because both glutamate (Glu) and glutamine (Gln) are very similar in their chemical structure, differing only in the addition of an amino group. Consequently, their resonance properties are similar, making it extremely difficult to detect them separately at low magnetic field strengths. Glutamate is mainly synthesized in neurons in the Krebs cycle, and once released at the synapse, is rapidly taken up by specific receptors in nearby astrocytes. There, glutamate is successively converted to glutamine, which then returns to glutamatergic and GABAergic neurons for re-synthesis of glutamate in GABAergic neurons. The continuous movement and transport of glutamate is significantly influenced by brain acidosis and, consequently, increased lactate production, which inhibits dopamine reuptake [26] and glutamate release [27]. This may represent a molecular mechanism explaining the endophenotypes of schizophrenia obtained by us in previous studies, for which levels in glutamine-glutamate transmission have been shown to be a key marker in differentiating individuals with schizophrenia [23].

In the present study, we attempt for the first time to elucidate a potential molecular mechanism of the previously observed changes related to NMDA receptor function, which undergoes lactate-dependent modulation in the brain of the astrocytic-neuronal shuttle that provides energy substrates (cAMP) between individual brain cells, ensuring the maintenance of synaptic plasticity responsible for, among other functions, memory [28]. Astrocytic increases in cAMP in specific areas of the brain induce lactate release [29], according to the direction of lactate movement dependent on concentration gradients produced by mitochondrial respiratory systems. Astrocytes, in this relationship, have an overriding function over glutamatergic down-regulated neurons, which are energy recipients. The maintenance of an appropriate concentration gradient is essential to maintain a continuous flow of lactate (from areas of high concentration to areas of low concentration), which can induce the expression of plasticity genes and so-called ‘long-term’ memory in neurons through modulation of NMDA receptor activity (via a mechanism of inhibition of the Erk1/2 signaling cascade) as a result of changes in the cellular redox state [30,31].

Although the presented mechanism of neuronal plasticity has only been demonstrated in rodent experiments, the assessment of these changes in humans may warrant further studies related to the development of effective therapies for the negative or cognitive symptoms of schizophrenia inextricably linked to the role of the astrocyte-neuronal transporter discussed above. These assumptions also go in the face of recent research findings that indicate the need to modify schizophrenia treatment regimens based on the efficacy of cognitive-behavioral therapies for hallucinations, delusions and negative symptoms as treatment based on protocols tailored to specific symptoms and patient sub-groups, depending on the stage of the illness, level of neurocognitive impairment and severity of the disorder, in order to achieve the desired end result [32].

Relying on the findings obtained in our earlier work, the present study assesses the association of the oral microbiome in previously studied clusters of schizophrenia patients with different brain lactate levels [23].

Several brain tissues, clinical and laboratory parameters were analyzed to find support for a concept related to the modulation of NMDA-R function dependent on the role of the lactate transporter in individuals with specific clinical phenotypes of schizophrenia.

This study aimed to examine the association of the oral microbiota with the effects of acid stress induced by an increase of brain lactate in schizophrenia patients, for whom negative and/or cognitive symptoms appeared to be predominant.

## 2. Materials and Methods

### 2.1. Participants of the Study

The study included 40 patients: 18 females and 22 males aged 14–35 years with acute multiform psychotic disorders meeting criteria for schizophrenia (F20 according to ICD-10) at 3-month follow-up.

### 2.2. Clinical Evaluation

Clinical and psychopathological methods, clinical scales (PANSS, Hamilton, Beck Depression Scale, Calgary Scale), psychological scales, catamnestic and clinical laboratory methods, and diagnostic imaging methods, nuclear magnetic resonance MRI and proton magnetic resonance spectroscopy (1H-MRS) in the frontal cortex (FC) and its subregions, were used.

The Positive and Negative Syndrome Scale (PANSS) [33] is a widely used tool that provides a comprehensive assessment of the psychopathology of schizophrenia, consisting of positive, negative and general psychopathological symptoms [33,34,35]. The PANSS is currently one of the most widely used psychometric tools for assessing symptoms of schizophrenia [36].

The Beck Depression Inventory—Second Edition (BDI-II) [37] is a self-report tool designed to obtain an index of the presence and severity of depressive symptoms over the past two weeks. It is one of the most widely used tools to measure the severity of depressive symptomatology [37]. The BDI-II features by a consistent design, with questions closely aligned to the criterion symptoms of depression. This design of the tool promotes high accuracy in diagnosing a depressive episode according to accepted diagnostic criteria and accuracy in differentiating from other mental illnesses. The scale consists of 21 items scored on a 0–3 scale, increasing according to the severity of symptoms. The level of depression is calculated from the number of points obtained after summing. The following breakdown was taken into account in the interpretation: 0–13 for indicating minimal depression or no depression; 14–19 for mild depression; 20–28 for moderate depression; and 29–63 for severe depression [37].

The psychometric properties of the Polish adaptation conducted by Łojek and Stańczak [38] showed excellent reliability, Cronbach’s α = 0.93 in the clinical group and α = 0.91 in the control group. Moreover, a satisfactory test-retest score (0.86) was achieved.

Although the BDI II as a self-report scale may provide more independent information about patients’ experience of depression in schizophrenia than interview-based assessments [39], it has been shown that patients with cognitive problems may find it difficult to complete questionnaires [40], and observable symptoms of depression may be overlooked by self-report [41]. Therefore, it was decided to additionally examine patients by a psychiatrist using the Hamilton Depression Rating Scale.

The Hamilton Rating Scale for Depression (HRSD) [42,43] is one of the most frequently used instruments by clinicians to measure the depth of depressive symptoms. It should be noted that it is also one of the most commonly used scales to measure depression in patients with schizophrenia [44]. The examination time using the HRDS is approximately 20–30 min. The patient is assessed on 17 items scored on a 5- or 3-point scale, giving a total score from zero to 52. Zimmerman et al. [45] established score ranges for the HRSD reflecting different levels of depression severity.

The Polish version of the HRSD, validated by Wiglusz et al. [46], was used in the conducted studies. The HRSD showed the best psychometric properties for a cut-off score of 11, with a sensitivity of 100%, specificity of 89.3%, the positive predictive value of 72.4% and negative predictive value of 100% [46], and the number of items used represented a total of 17 items. The version used includes items on somatic thoughts and suicidal thoughts, but lacks items that would be necessary for the diagnosis of a major depressive episode (for example, sleep difficulties or weight increase). Each item was scored on a 3- or 5-point scale and summed to give a total score.

The Calgary Depression Scale for Schizophrenia (CDSS) [47], (Polish adaptation [48]) is the only scale specifically designed and developed to assess the level of depressive symptoms in patients with schizophrenia spectrum disorders. Importantly, the scale distinguishes depressive symptoms from positive, negative and extrapyramidal symptoms [49]. The SDSS can be used in patients with exacerbations of psychosis, as well as in remission. The scale consists of 9 items, which the clinician scores from 0 (absence of symptom) to 3 (maximum symptom severity). A score is obtained by adding up the points for each item. The range of scores is 0 to 27. A higher total score reflects more severe depression. The time frame refers to the previous 2 weeks. Studies of the relevance and reliability of the scale have shown its ease of use, adequate univariate structure and good internal consistency, high inter-rater reliability and satisfactory external validity and specificity [40,49,50].

State and Trait Anxiety Inventory (STAI) The State and Trait Anxiety Inventory (STAI) of Spielberger et al. [51] is used to assess anxiety as a state and as a trait [52].

The Childhood Trauma Questionnaire (CTQ) is a retrospective self-report scale assessing the level of exposure to trauma experienced in childhood and adolescence, developed by David Bernstein and colleagues [53,54,55]. The CTQ measures five types of trauma: abuse (emotional (EN), physical (PN), sexual (SA)), neglect (emotional (EN) and physical (PN) and three items on minimizing/denial tendencies. The test was adapted into Polish with the permission of the authors [56]. The CTQ was administered to patients at the time of clinical stabilization within 3 months of hospital admission.

The Premorbid Adjustment Scale (PAS) [57] is one of the most widely used measures of pre-morbid adjustment in schizophrenia populations [57]. The PAS scale was designed to measure pre-morbid functioning from a developmental perspective, based on the premise that good pre-morbid adaptation is the key to achieving age-appropriate developmental and social milestones. Using the PAS questionnaire [57,58], functioning was analyzed in four developmental phases: childhood (up to 11 years), early adolescence (12–15 years), late adolescence (16–18 years) and adulthood (≥19 years), as well as in five domains: sociability/social withdrawal, peer relationships, school achievement, adaptation to socio-sexual functioning excluding childhood. In addition to the four developmental scales, the PAS includes an overall scale that assesses factors such as the individual’s achieved level of best functioning, as well as items related to onset characteristics, energy level, education and self-efficacy.

The Montreal cognitive assessment scale (MoCA) [59] was designed as a tool to detect cognitive dysfunction, which is an underlying and persistent feature of schizophrenia [60]. It can be used to assess different areas of cognitive functioning: attention and concentration, executive functions, memory, language, visual-motor skills, conceptual thinking, calculus and orientation. Total scores range from 0 to 30, with higher scores indicating better cognitive functioning. A score of 26 and above is interpreted as normal. The MoCA has been shown to be a useful tool for monitoring cognitive function in patients with schizophrenia [61,62,63].

Brain imaging studies

MR spectroscopy (MRS—magnetic resonance spectroscopy) spectrum is described by two axes: the vertical axis represents the signal intensity or relative concentration of different brain metabolites, and the horizontal axis is used to determine the type of metabolite based on its chemical shift. The nature of the chemical shift effect is to induce a change in resonance frequency for nuclei of the same type located in different chemical groups. The resulting frequency difference can be used to identify the presence of important chemical compounds. By monitoring individual peaks, MRS can provide a qualitative and/or quantitative analysis of a range of metabolites in the brain if reference is applied to a known concentration of metabolites at a given magnetic field strength [64]. Furthermore, magnetic resonance spectroscopy provides information on the altered flow rate of various metabolic pathways, including those related to the alanine-lactate transporter and glutamatergic transmission, in health and in various diseases. Assessment of the metabolic flow rate of the LAC/ALA and/or Glx cycle appears to be crucial in neuropsychiatric diseases, as it directly reflects the production and changes of these brain metabolites in real time [65].

The study protocol was approved by the UJ Bioethics Committee (Consents number: 1072.6120.152.2019 of 27 June 2019, 1072.6120.17.2020 of 27 February 2020 and 1072.6120.252.2021 of 17 January 2022). All study participants gave informed written and verbal consent prior to head scanning, which was approved by the Bioethics Committee at the Jagiellonian University Medical College. Patients and volunteers in the control group first underwent imaging of the brain to exclude possible morphological changes in the central nervous system. The imaging study was performed in T1-, T2-weighted, FLAIR and DWI sequences in transverse, frontal or sagittal planes in layers of 3 mm–5 mm. The standard protocol for MR imaging of the head without contrast in force at the diagnostic laboratory of the University Hospital in Krakow was used.

The next imaging step was single (localized) voxel spectroscopy (SVS) and SAGE 7.0 software (Plano, TX, USA). From each participant, a three-plane image (VOI) of the order of 6–8 cm^3^ was acquired from the prefrontal cortex (DLPFC), anterior cingulate cortex (ACC), parallel to and above the dorsal anterior surface of the corpus callosum, centered on the interhemispheric fissure. Each spectroscopic spectrum was subjected to qualitative and quantitative analysis. MRS spectra were acquired at 2 min and 12 s in a Siemens MR 1.5T scanner, Siemens Healthcare GmbH, 91052 Erlangen, Germany (with nominal proton frequency = 123 MHz; General Electric Healthcare, Milwaukee, WI, USA) using an 8-channel phased-array and receive-only head coil, using the Point-Resolved Spectroscopy Sequence (PRESS) with water suppression in the CHESS (CHEmical shift Selective Imaging Sequence) [66]. The SPECIAL sequence was used for acquisition without water suppression as the reference signal for all three sequences [67].

MEGA-PRESS acquisition [68] using frequency-selective editing pulses at δ = 1.33 ppm was used to edit the lactate-derived signal (LAC), and frequency-selective editing pulses at δ = 1.48 ppm were used for alanine (ALA) (Figure 1).

Automatic pads were used to obtain good quality spectra. Magnetic resonance spectroscopy acquisition parameters are described in detail in our previous work (MRS) [23]. Spectroscopic analysis was performed using SAGE 7.0 software (Plano, TX, USA). The analysis was performed in the following steps: (1) zero-filling (always to the power of 2), (2) Fourier transformation, (3) baseline correction, (4) automatic phase correction and (5) curve fitting, performed based on the Gaussian shape to calculate the peak area.

### 2.3. Preparation and Analysis of Blood Samples by Liquid Chromatography Coupled to Tandem Mass Spectrometry (LC-ESI-MS/MS)

Blood was collected from all study participants from the median vein of the elbow using a closed blood collection system (SARSTEDT, Germany) with a serum activator. Blood samples were left to activate coagulation for approximately 10–20 min and then centrifuged at 1200× *g* for 15 min. The supernatant (serum) was then stored at −80 °C until analysis.

Serum samples were thawed and analyzed by LC-ESI-MS/MS, using the method described by Li et al. with an in-house modification shown below [69].

Blood concentrations of selected amino acids were analyzed according to the protocol: 4 µL of internal standard, which was a methanolic solution containing isotopic derivatives of the compounds under study (alanine-d4: ALA-d4, serotonin-d4: SER-d4 and glutamic acid-C13: GLUT-C13) at a concentration of c = 500 µg/mL, was added to a 100 µL aliquot of the previously prepared blood serum sample and vortexed for 10 s. This mixture was then deproteinized with 100 µL acetonitrile (1:1, *v/v*) and vortexed for 30 s. The samples were centrifuged at 8000 rpm for 10 min at 15 °C. A 100 µL aliquot of the supernatant was placed in chromatographic tubes and analyzed chromatographically.

Chromatographic separation was carried out using a WATERS ACQUITY UPLC^®^ H-Class system liquid chromatograph (Waters Corporation, Milford, MA, USA). Samples were separated on a ZIC^®^-HILIC column (5 µm, 200 Å, 150 mm × 2.1 mm; Merck, Darmstadt, Germany) and placed in a thermostat at 40 °C. A total of 0.1% aqueous r-rice of HCOOH (phase A) and 0.1% HCOOH in acetonitrile (phase B) were used as mobile phases, the phase flow rate was 0.6 mL/min.

A gradient separation scheme was used: 0–0.2 min, 5.0% isocratic gradient (A); 0.2–1.5 min, 5–55% linear gradient (A); 1.5–3.1 min, 55% linear gradient (A); 3.1–4.5 min, 55.0–5.0% linear gradient (A); 4.5–6 min, 5.0% isocratic gradient (A); flow rate: 0.6 mL/min. The volume fed to the column was 4 µL. The duration of analysis for a single sample was 6 min. Retention times for individual compounds were for alanine (ALA; ALA-d4)–2.44 min, serotonin (SER; SER-d4)–2.35 min, glutamic acid (GLUT; GLUT-C13)–2.40 min and lactate (ML)–0.99 min.

Quantitative analysis was performed using a Xevo TQ-S^®^ quadrupole mass spectrometer (Applied Biosystems MDS Sciex; Concord, ON, Canada) with electrospray ionisation (ESI) option in positive ionisation mode. Ion source parameters: sputtering voltage (IS): 5500 V; spray gas (gas 1): 30 psi; turbo gas (gas 2): 20 psi; the temperature of the heated nebuliser (TEM): 550 °C; shielding gas (CUR): 30 psi. For quantitative evaluation, the analysis was carried out in selected reaction monitoring mode (MRM). Selected ion pairs were monitored with values of *m*/*z* = 90.09/44.9 for ALA, *m*/*z* = 93.93/47.97 for ALA-d4, *m*/*z* = 148.0/84.0 for GLUT, *m*/*z* = 153.0/89.0 for GLUT-C13, *m*/*z* = 177.0/119.0 for SER and *m*/*z* = 182.83/119.76 for SER-d4. For lactate (LAC) analysis, a negative ionisation mode was used. An ion pair with *m*/*z* = 88.84/43.948 (lactate, LAC) was monitored. Data were processed using MassLynx V4.2 software, Waters Corporation, Milford, MA, USA. Concentration of analytes was calculated using calibration standard curves constructed by linear regression analysis for peak area versus concentration (Figure 2).

### 2.4. Isolation of Oral Microorganisms and Identification by Matrix-Assisted Laser Desorption Ionization (MALDI-TOF MS)

Using a sterile cotton swab, swabs were taken from the buccal and dorsal tongue mucosa and saliva samples according to methods developed previously [70,71,72,73]. The choice of the above-mentioned oral locations for swab collection is dictated by the fact that the microorganisms of the tongue constitute a relatively stable, resident oral microflora [74]. The choice of saliva, on the other hand, is dictated by the fact that it contains mainly planktonic forms that mediate oral dysbiosis through, among other things, the contribution of metabolic activity products of microorganisms accumulated in both biofilms and the liquid phase. Additional support for this approach is provided by the high species diversity, taking into account both the permanent oral inhabitants responsible for the development of local diseases, such as caries, halitosis or periodontitis (for which the aetiological factors of these diseases are crucial in the development of neuropsychiatric diseases, such as Alzheimer’s disease, Autism or Schizophrenia), as well as incoming and/or mediated exchanges between oral biofilms and transitional (dormant) forms, which, under favorable conditions, acquire virulence characteristics initiating the development of the above-mentioned diseases [75,76,77,78].

Samples were transported within 2 h to the laboratory in 1 mL sterile saline pH 7.0 (PBS) at room temperature. Samples were homogenized by gentle shaking and sonication for 30 s at room temperature. Microbiological analysis was performed by quantitative and qualitative assessment of the oral microbiota. For this purpose, the method of inoculating serial dilutions onto solid media, both selective and non-selective, was used. Serial dilutions of the stock solution were prepared in sterile saline. Then, 100 µL samples of the prepared dilutions were inoculated onto 5% blood agar plates (Columbia medium) and consistently onto selective-nonselective media, which allow the growth of specific bacterial species, while inhibiting the growth of others. The following selective media were used: CHROMagar TM, a chromogenic agar supporting *Candida* and yeast growth; BD MacConkey II selective agar for the isolation and differentiation of *Enterobacteriaceae*; BD™ LBS agar for the isolation and differentiation of *Lactobacillus*; chocolate agar plates for challenging Gram-negative bacteria; MacConkey agar; OPA agar plates. Inoculated media were incubated under microaerophilic conditions in the presence of 5% CO_2_ at 37 °C for 24–48 h.

Isolated microorganism species were analyzed by mass spectrometry (MS) using a MALDI-TOF MS Biotyper 3.0 Microflex system (Bruker Biotyper; Bruker Daltonics, Bremen, Germany) connected to a database of microorganism profiles.

The MALDI-TOF method was used in the ongoing study due to the limited cost of conducting the study compared to the use of 16 s RNA sequencing, which is still a more expensive technology and requires bioinformatics tools to analyze the results obtained. Cultures in combination with MALDI-TOF are well established methods used by us in the identification of microorganisms from clinical specimens [72]. Matrix-assisted laser desorption time-of-flight mass spectrometry (MALDI-TOF MS) is a standard method for clinical microbiology and is the method of choice for identifying microorganisms in clinical laboratories. This method is routinely used for clinical diagnostic testing, which supported the research conducted by the authors of this manuscript [79,80,81,82,83]. In addition, despite some limitations, culture combined with MALDI-TOF MS can play a crucial role in the analysis of certain bacterial species, among others *Enterobacteriaceae*, which can account for up to 67% of the total number of isolated microbial species that are cultured, but not identified by 16 S rRNA NGS in the test material [84].

Before the actual identification, a pre-extraction (using ethanol and formic acid) was carried out, which is known to be superior to the direct method of identifying colonies of oral bacterial species, i.e., lactobacilli [85]. Two microliters of the extract thus prepared were placed on a metal plate and allowed to dry at room temperature. During the measurement, the metal plate with the applied samples was placed in the chamber of the MALDI Biotyper apparatus and exposed to a laser. Under desorption and ionization of bacterial proteins, the ionized peptides were accelerated in a strong electric field, and the TOF ion transit time was measured. Based on the resulting distribution of peptides according to molecular weight, charge and variable time of flight, the MALDI-TOF MS system automatically generated spectrometric spectra of peaks corresponding to ions with variable mass versus charge and analyzed the number, intensity and correlation of peaks and compared the studied spectrum with reference spectra of microorganisms from the database. The spectra were recorded in the range 2000–20,000 Da at the maximum laser frequency. The raw spectra were then analyzed automatically using the MALDI software package Biotyper 3.1 (Bruker Daltonics GmbH, Bremen, Germany, Biotyper^®^ database version renewed 2020) (Figure 3). Logarithmic results ranging from 0 to 3.00 were obtained, which were interpreted according to the manufacturer’s recommended criteria as ≥1.7 as a reliable identification for the species, while <1.7 as a reliable identification at the genus level.

### 2.5. Statistical Analysis

Statistical analysis was carried out using the IBM SPSS Statistics 25 package. Spearman’s correlation analysis checked for a statistically significant relationship between the analyzed variables. The analysis with the Kruskal–Wallis test allowed us to check whether there were statistically significant differences between the 3 groups of people, distinguished on the basis of severity of depressive disorders. If statistically significant differences were present, the Games–Howell post-hoc test was used to test for them. Using k-means cluster analysis, clusters of individuals were extracted on the basis of the T 2 scale, and then using the Mann–Whitney U test, it was tested whether they differed in terms of the analyzed variables. Correlation analysis was performed in order to reveal significant correlations between continuous variables. In the regression analysis, two statistically significant predictors were included to see how they influence, in their mutual presence, the Hamilton scale scores obtained by the subjects. A *p*-value < 0.05 was taken as the statistically significant level.

## 3. Results

### 3.1. Cluster Analysis—Unsupervised Clustering

Using cluster analysis on the basis of the primary endpoint of improvement in total score on the Positive and Negative Syndrome Scale (PANSS-T), two equal clusters of individuals were created. Each cluster had a population size of 20. Descriptive statistics for the T Scale in the identified clusters are provided in Table 1 and Appendix A. The second cluster of individuals was characterized by statistically significantly lower T-scale scores in relation to the first, as confirmed by the effect size (Table 1).

Among all analyzed numerical variables from: clinical assessment, laboratory assessment, including microbiology, and brain variables, statistically significant differences between separate clusters in the group of people with schizophrenia concerned brain variables, i.e., lactate and alanine levels in the prefrontal cortex, a brain area located in the anterior cingulate cortex (ACC): ACC_LAC and ACC_ALA, with *p* values of 0.03 and 0.04, respectively (Appendix A). The similarities in classification patterns in these two patient clusters suggest that their typology is relatively stable over the course of the disease, which may address the heterogeneity of brain lesions at different stages of the disease and objectively provide tools for more effective treatment of selected patient subgroups. Those with higher total scores on the Positive and Negative Syndrome Scale (PANSS-Total) (cluster 1) had statistically significantly lower LAC lactate concentrations compared to the other cluster with higher LAC concentrations. No statistically significant differences were found for the other variables analyzed.

### 3.2. Evaluation of the Relationship between Oral Microbiota and Subjective Symptoms, Brain Metabolic Activity and Biochemical Markers of People with Schizophrenia

The species composition and percentage of oral microbial species isolated are presented in Figure 4. The analysis of the relationships between oral microbiota, subjective symptoms, brain metabolic activity and peripheral markers of schizophrenia patients is presented in Table 2.

### 3.3. Assessment of the Association of Clinical Status with the Oral Microbiota of Patients with Schizophrenia

The total score on the Positive and Negative Syndrome Scale (PANSS-T) showed a statistically significant association with the following variables on oral microbiome analysis: *Gemella haemolysans* (r_s_ = 0.34; *p* = 0.04), *Leptotrichia* sp., (r_s_ = 0.32; *p* = 0.049), *Streptococcus sobrinus* (r_s_ = 0.34; *p* = 0.04), *Prevotella nigrescens* (r_s_ = 0.37; *p* = 0.02). The association for *Prevotella nigrescens* was found to be the most significant (Figure 5). As for the other variables entering into a statistically significant association, there are only single individuals for which the score is higher than zero.

The assessment of the associations with the microbiological data revealed statistical significance for a number of variables, which are highlighted in Table 2. Several species appeared to have a stronger association with the Hamilton Scale than with the Total (T) score on the PANSS scale: *Lactobacillus rhamnosus*, *Streptococcus salivarius*, *Streptococcus parasanguinis*, *Lactobacillus acidophilus* correlated negatively, i.e., the higher the score on the Hamilton Scale (greater severity of depression), the lower the number of microbial species tested was observed. Species that showed strong positive correlations with the Hamilton scale were: *Neisseria macacae*, *Neisseria mucosa*, *Rothia mucilaginosa*, *Scardovia wiggsiae, Streptococcus mutans*, *Staphylococcus epidermidis*, *Veilonella* sp., *Prevotella nigrescens*, *Staphylococcus aureus*.

The statistically significant associations are stronger when compared to the PANSS-T score and other tools used to assess the clinical status of patients with schizophrenia. The associations between *Lactobacillus rhamnosus* and *Streptococcus salivarius* with the subjects’ Hamilton scale score are negative and strong (Figure 6). In contrast, the association between *Prevotella nigrescens* and the Hamilton scale score obtained by the subjects is strongly positive (Figure 7).

### 3.4. Evaluation of the Relationship between Oral Microbiota and Subjective Symptoms, Brain Metabolic Activity and Biochemical Markers of People with Schizophrenia

Two variables were included in the regression analysis for the prediction of the Hamilton score, i.e., P.ngs, because they did not correlate in a statistically significant way with the above-mentioned variables, and S.par, because of the strongest association and a large number of individuals with a specific outcome (>0). More predictors were not included in the regression analysis in order not to fall below 15–20 individuals per predictor and because of the statistically significant correlations between potential predictors, which could contribute to erroneous results in the regression analysis. The model composed of these predictors proved to be statistically significant. F (2;36) = 20.71; *p* < 0.001. S.par (Beta = −0.62; *p* < 0.001) was found to be a stronger predictor of Hamilton scores than P.ngs (Beta = 0.26; *p* = 0.03). The model composed of these 2 variables explains 51% of the variance in the variable Hamilton score.

The assessment of depression severity showed a statistically significant correlation with species of oral microbiota, viz.: *Bacillus circulans*. B.c. (r_s_ = −0.37; *p* = 0.02); *Leptotrichia sp.* L.sp (r_s_ = 0.38; *p* = 0.02); *Neisseria mucosa*. N.mu (r_s_ = 0.4; *p* = 0.01); *Scardovia wiggsiae* Sc.w. (r_s_ = 0.4; *p* = 0.01); *Lactobacillus acidophilus*. L.ac. (r_s_ = −0.52; *p* = 0.001); *Staphylococcus epidermidis* S.e (r_s_ = 0.56; *p* = 0.001); *Streptococcus mutans* S.mu (r_s_ = 0.36; *p* = 0.03); *Streptococcus parasanguinis* S.par (r_s_ = −0.72; *p* < 0.001); *Streptococcus salivarius* S.sal (r_s_ = −0.56; *p* < 0.001). Three of the above-mentioned oral microbiota compounds are characterized by the required group size, thus these were further analyzed.

Statistically significant associations between oral microbiota species and depression status are shown in Table 3. The different depressive disorder severity groups differed in 3 parameters: S.sal, H = 12.71, *p* = 0.002, eta^2^ = 0.3; S.par, H = 23.85, *p* < 0.001, eta^2^ = 0.61; S.e, H = 12.05, *p* = 0.002, eta^2^ = 0.28. Pairwise comparisons showed that those with the highest severity of depressive disorders scored lower on *Streptococcus salivarius* than S.sal for mild disorders (*p* = 0.003) and no depression (*p* = 0.02). Concerning *Streptococcus parasanguinis*, S.par, those with no depression scored higher on this variable with mild disorders (*p* = 0.003) and moderate disorders (*p* = 0.001). Regarding *Staphylococcus epidermidis*, S.e, patients with no depression scored lower on this variable (*p* = 0.004) compared to the most severe depressive disorders.

## 4. Discussion

### 4.1. Cluster Analysis—Unsupervised Clustering

The project presented here is a continuation of a study we started earlier, in which the effect of altering the oral microflora on N-methyl d-aspartate (NMDA) receptor function associated with the alanine-lactate transporter was found to be central to glutamatergic neurotransmission. In our preliminary study, diagnosis and central cluster effects related to glutamatergic transmission overlapped in the anterior cingulate region of the ACC prefrontal cortex [23].

In the model created during this project, among the numerical variables related to clinical assessment, laboratory evaluation and brain variables, which were included in the unsupervised clustering, statistically significant differences in the schizophrenia group were related to brain variables, i.e., lactate and alanine levels in the anterior cingulate cortex. Subjects from cluster 1, with a significantly higher total score on the Positive and Negative Syndrome Scale (PANSS-T, Table 1), had statistically significantly lower lactate levels compared to the second cluster (with a lower total score) with higher lactate levels in the ACC area (LAC_ACC) (Figure 1), which may provide an explanation for our previous observations, in which endophenotype I was characterized by higher glutamatergic transmission, while endophenotype II was characterized by individuals with lower glutamate levels [23].

The concept of N-methyl d-aspartate (NMDA) receptor function indicates that underactivation of these receptors in neurons mediating the release of gamma-aminobutyric acid (GABA, the main inhibitory neurotransmitter) becomes the main cause of compensatory-increased presynaptic glutamate (Glu) release. This was confirmed in our preliminary studies, where changes in glutamatergic (Glx) transmission levels were found to be key markers to significantly differentiate patients with schizophrenia. Indeed, the results observed by us previously indicated that Endophenotype I was characterized by increased glutaminergic transmission, and Endophenotype II showed decreased glutaminergic activity. Changes in Endophenotype I showed a stronger association with negative and positive symptoms of patients with schizophrenia compared to patients of Endophenotype II with reduced glutaminergic activity [23]. Our results find strong support in the NMDA receptor theory, which posits that GABA synthesis and release are reduced [86] during paradoxically increased glutamate output. Consistently, when there is no negative feedback between excitatory and inhibitory neurotransmission, then dopamine levels in the brain increase, leading to the appearance of positive and negative or cognitive symptoms of schizophrenia [87]. One of the actions of NMDA-R antagonists is to reduce the excitation of fast-excitatory interneurons—mediating neurons—resulting in the disinhibition of pyramidal cells. Overactive pyramidal cells, in turn, can induce a hyperdopaminergic state that causes psychosis. These changes become the cause of further abnormalities in homeostatic systems, leading to NMDA-R inactivity, which reduces burster and can induce negative and cognitive symptoms of schizophrenia [88,89].

The current project can hypothesize an explanation for our previously obtained results by showing that the continuous movement and transport of glutamate may be significantly affected by brain acidosis resulting from increased lactate production in the brain, which inhibits dopamine reuptake [26] and glutamate release [27]. On the one hand, an increase in lactic acid levels may suggest a primary metabolic disorder in schizophrenia, although it may also represent an effect secondary to prior antipsychotic treatment [90]. Indeed, commonly used antipsychotics, i.e., clozapine or haloperidol, mainly target dopamine (D2 receptors), effectively alleviating the positive symptoms of schizophrenia. Negative symptoms or cognitive impairment related to neurotransmitters, i.e., glutamate or γ-aminobutyric acid (GABA) controlled by NMDA signalling, are particularly relevant in drug-resistant schizophrenia, contributing to energy deficits in this disorder [91,92]. Our previous studies indicate that the pathophysiology of individuals with schizophrenia endophenotypes is closely linked to deficits in anterior cingulate cortex (ACC) bioenergetic function. The involvement of glutamatergic transmission in shaping the endophenotypes of schizophrenia has shown that, despite antipsychotic treatment of the positive psychotic symptoms present, the therapeutic effect of the drugs used is not fully apparent for the full profile of schizophrenia symptoms, such as negative and cognitive symptoms [23].

An explanation for our previous findings is the concept of an alanine-lactate shuttle supplying energy substrates (cAMP) between individual brain cells, ensuring the maintenance of synaptic plasticity, according to the theory proposed by Waagepetersen et al. [93]. This model assumes that the vesicular release of glutamate in neurons by N-methyl-D-aspartate (NMDA) is mediated by the lactate-alanine transporter, from which lactate is released via astrocytes, and the released glutamate is absorbed [94]. Astrocytic increases in cAMP in specific areas of the brain induce the release of lactate [29,94] according to the direction of lactate movement, dependent on the concentration gradients produced by mitochondrial respiratory systems. Astrocytes, in this relationship, have an overriding function over glutamatergic neurons, which are energy recipients. The maintenance of an appropriate concentration gradient is essential to maintain a continuous flow of lactate (from areas of high concentration to areas of low concentration), which can induce the expression of plasticity and so-called ‘long-term’ memory genes in neurons through modulation of NMDA receptor activity as a result of changes in the cellular redox state [30,31].

It is known that glutamate cannot cross the blood–brain barrier, hence its synthesis takes place in the brain. The glutamate–glutamine cycle takes place between glutamatergic neurons and astrocytes compensating for the neuronal loss of the neurotransmitter by transferring the glutamate precursor glutamine [95]. Astrocytes are the main producers of glutamine in the central nervous system (CNS) via glutamine synthetase (GS). The reactions involving alanine to produce glutamate take place in the brain via alanine aminotransferase (ALAT) and branched-chain amino acids, i.a., transaminases (BCAT) [96,97]. Glutamate is directed to glutamatergic neurons in partial exchange for alanine. In co-cultures of neurons and astrocytes, there is a threefold greater release of glutamine than basal glutamate release. This is related to astrocyte-derived alanine being used to synthesize glutamine, which is released from neurons into astrocytes in exchange for lactate. Lactate transported into astrocytes provides a substrate for pyruvate synthesis, which shifts the NADH/NAD ratio, triggering activity associated with increased glutamate transport into astrocyte mitochondria, which produce increased amounts of ammonia required for glutamine synthesis. The changes in alanine and lactate concentrations in ACC likely observed in the studies presented here are closely related to altered brain energy metabolism in schizophrenia, which consequently leads to lactate accumulation and increased acidity in specific brain areas. Reduced aerobic metabolism leads to shifts away from processes, i.e., tricarboxylic acid (TCA) cycle and oxidative phosphorylation, towards a relatively anaerobic metabolism based on glycolysis as the main energy source [4]. Most ongoing studies point to an increase in the products of glycolysis, mainly lactate, together with a decrease in TCA cycle intermediates [64] in the brain as key to the pathomechanism of schizophrenia. Lactates are one of the most important modulators of acid-sensing ion channels (ASICs), which are activated by extracellular acidification. These channels are sensors for detecting initial rapid changes in pH, involved in, among other things, neuronal death after ischemic stroke [97]. They belong to voltage-gated cation channels, whose role is related to enhancing nerve conduction by relieving the extracellular Ca^2+^ block in response to neuronal injury. The functional effects of cerebral acidosis are considered to be consequences of ASIC channel modulation [98]. An excellent example in support of this theory is ischaemic stroke, in the course of which it takes several hours for acidosis and activation of ASIC1a channels to cause irreversible changes in the brain—from the ischaemic core to the ischaemic regions of the hemisphere [99].

Both acidotoxicity and excitotoxicity of the glutamate-modulating NMDA receptor function are the two main mechanisms underlying death in the ischaemic brain. In ischaemic stroke, the duration of protection associated with the blockade of ASIC1a channels is 5 h [100,101], which is much longer than for NMDA receptor function. In addition, animals lacking ASIC channels (ASIC1-/-) show no obvious defects in physiological function [102], potentially providing hope for novel clinical therapies for both ischaemic stroke and the need to modify schizophrenia treatment regimens tailored to specific symptoms and patient subgroups.

The different levels of lactate and alanine in ACC presented by us in this study, differentiating individuals from schizophrenia endophenotypes, provide support for new therapeutic targets related to the function of ASIC1a channels as key to acidosis during neuronal damage in this disease. Assessment of neuronal function via ASIC1a channels and ALA-LAC-dependent NMDA regulatory mechanisms may support novel therapies for schizophrenia, for which acidosis is a differentiating feature of individuals with schizophrenia endophenotypes.

Our findings are unprecedented and in line with the recently published results of the whole exome of people with schizophrenia. This study demonstrated a key role in genes associated with postsynaptic, as well as presynaptic, pathology. Genes encoding ion receptors and channels, including voltage-gated calcium and chloride channels (CACNA1C, CLCN3) and the ligand-gated NMDA receptor subunit (GRIN2A), which is regulated by NMDA transmission and regulates NMDA receptor abundance, were cited as key factors in the development of schizophrenia [101].

Our results focus on neuroimaging markers as key to phenotypic changes in molecular and cellular therapeutic targets associated with the treatment of schizophrenia [102]. These biomarkers, which are directly related to the pathomechanism of the disease, may provide a factual measure of the pathophysiological underpinnings of the disease process, serving to assess direct or surrogate to clinical assessment endpoints expressed on the total PANSS scale [103].

### 4.2. Assessment of the Association of Oral Microbiota with Clinical Status and Other Biochemical Parameters in Patients with Schizophrenia

The changes in brain lactate levels observed in the present study may also result from metabolism by the gastrointestinal microflora, one effect of which is to promote anxiety-like behavior [104].

The aetiopathogenesis of schizophrenia is multifactorial. The human microbiota is one of the etiological factors of this disease [105,106,107]. Therefore, it is reasonable to define the role of microorganisms colonizing particular ecological niches of the human body in the initiation and progression of schizophrenia [108].

Hypotheses suggest that the oral microbiota communicates with the brain through systemic inflammatory pathways triggered by selected oral disease pathogens [109,110]. A growing body of evidence suggests that the oral microbiota contributes to a number of systemic conditions, such as hyperglycaemia [111], hypertension [112], cardiovascular disease [113], obesity [114], Alzheimer’s disease, depression and autism spectrum disorders [78,115]. Common mechanisms include chronic inflammatory conditions, i.e., caries, gingivitis and periodontitis, caused by the microbiota involved in causing oral disease [116,117]. An example is the low-grade inflammation typical of periodontitis. whose effect on endothelial function is associated with arterial stiffness and generation of elevated blood pressure [118]. In this sense, reducing inflammation caused by oral bacteria is also effective in improving endothelial function in the long term.

As we have shown in our previous studies, schizophrenia is also characterized by chronic low-grade inflammation with significant effects on peripheral microcirculation and endothelial function [119,120,121], with concomitant effects on cerebral microcirculation (which was associated with reduced movement of micronutrient molecules from astrocytes to neurons, as measured by the diffusion tensor in the right frontal lobe) [122]. Presumably, bacteria residing in the human gastrointestinal tract and their metabolites play an important role in these processes. Another immunological process, the redox status, may favor certain bacteria over others in schizophrenia, as demonstrated by the negative correlation observed in our previous studies between the total antioxidant potential of saliva, triiodothyronine (T3) and negative symptoms as an expression of compensation of saliva antioxidant systems against glutamate-induced neuronal toxicity [122]. The oral microbiota shows a number of disease-specific associations with clinical assessment indicators, oral microbiota parameters and peripheral indicators, which may account for the metabolic activity of selected oral microbiota species [Table 2] in schizophrenia.

As the oral cavity is the origin of the gastrointestinal tract, it may represent an extension of the microbial gut–brain axis into the microbial oral–brain axis. Oral bacteria can communicate with the brain via a number of routes, including via the small blood vessel pathway, via the olfactory tract and systemic circulation, via the leaky gut and the vagus nerve and via alterations in the blood–brain barrier, being the cause of the ultrastructural and redox imbalance of the brain via, among others, short-chain fatty acids and a number of soluble inflammatory mediators, in particular brain-derived neurotrophic factor, N-methyl-d-aspartate receptor subunit 2B or the serotonin transporter [123,124,125,126,127,128,129,130].

Most research on the human microbiome in schizophrenia has so far been conducted about changes in the gut [108,109,131]. Results from animal studies show that the gut microbiome has a significant impact on the onset of schizophrenia, including cognitive function and behavior [132,133]. Mice receiving fecal transplants from people with schizophrenia showed lower glutamate levels and higher glutamine levels in the hippocampi, combined with induction of schizophrenia-typical behavior [134]. Additional observations indicate the influence of gut bacterial metabolic products on the occurrence of more or less severe clinical symptoms of the illness [135]. There are single papers that indicate a role for oral and pharyngeal bacteria [110,112,136] in the development of the disease.

Of the more than 700 species of bacteria colonizing the oral cavity, three have been linked to cerebrovascular disease: *S. mutans*, the aetiological agent of dental caries [137]; *Porphyromonas endodontalis,* involved in endodontic infections; and *P. gingivalis,* associated with gingivitis and periodontitis [138,139]. Higher levels of lactic acid-producing bacteria and higher levels of bacteria associated with glutamate and GABA metabolism, relative to lower levels of short-chain fatty acid-producing bacterial genera (*e.g.*, butyrate), were singled out in a systematic review of the composition of the gut microbiota as key to the development of serious psychiatric disorders, including schizophrenia [140]. In addition, many salivary bacterial species were strongly associated with alanine synthesis pathways, aspartate metabolism and glutamate metabolism in patients in a first psychotic episode [141], which supports the strong positive correlations obtained in this study between oral microbiota species, i.e., *Prevotella nigrescens*, *Veilonella* spp., *Neisseria* spp. and *Leptotrichia* sp. *Rothia mucilaginosa,* and peripheral blood alanine levels (Table 2). The results also provide support for the claim that communication between the oral microbiota and the brain may take place, among other things, through pathways for the synthesis of amino acids, such as alanine or glutamate, which may represent both substrates and products of the metabolism of these bacteria.

These facts determine that alterations in the gastrointestinal microbiota, including the oral cavity, may represent a strategy for disease diagnosis and therapeutic intervention in the treatment or prevention of schizophrenia.

This became the rationale for the present study, in which we wished to determine the association of the oral microbiota with the effects of an acid stress mechanism induced by an increase in brain lactate, which may represent both a product and a substrate utilised by oral microflora bacteria. Our observations shed new light on the influence of selected elements of the human microbiota in shaping the endophenotypes of schizophrenia, opening up new possibilities for therapeutic interventions to reduce the burden of untreatable negative and/or cognitive symptoms of the illness.

Increased lactate levels in the brain carry shifts in the microbiota in favour of lactate-utilising bacterial genera, e.g., *Escherichia*/*Shigella*. *Veillonella* is somehow a compensatory mechanism in response to increased lactate production in the brain. The effect is compounded by shifts in the microflora in favour of lactate-metabolising bacteria (i.e., *Veillonella* or *Actinomyces*), which was also supported in our study, in which a positive correlation was observed between *Veilonella* spp. isolated from the mouth and peripheral blood lactate levels (Table 2, r_s_ = 0.344; *p* = 0.032) or *Actinomyces* levels and anterior cingulate ACC_LAC lactate levels (Table 2, r_s_ = 0.360; *p* = 0.024). This is related to the fact that lactate, which is a metabolic product for one bacterial species (*Actinomyces*) (observed correlation between *Actinomyces graevenitzii* and *Veilonella* spp.; Table 2. r_s_ = 0.371; *p* = 0.020), is at the same time a nutrient substrate for other species (*Veillonella*), which use it, among other things, to produce propionic and acetic acid during hydrogen production [142]. In turn, an excess of propionate has a documented effect associated with an increase in depression-like symptoms in animal models of depression [143]. In addition, there is a hypothesis that the by-product of lactate metabolism, hydrogen, may accompany species shifts involving the reduction of sulphate-producing bacteria that produce hydrogen sulphide, methane and acetate [144]. Pathways related to methane metabolism and oxidation have also found their place in the pathophysiology of psychiatric disorders [20]. Shifts or interactions between components of the microbiome may influence the pathophysiology of psychiatric disorders through the accumulation of lactate in the brain, which provides support for the so-called glucose alternative neurobiological pathway hypothesis (related to neuronal activation) using lactate as a primary energy source [22].

Interactions between the microbiota and the human brain take place through direct and indirect pathways, through which the microbiota can influence the CNS, cognitive function, mood and vice versa [145]. This happens, among others, through the tryptophan-kynurenine pathway, which is altered in patients with depression, schizophrenia or paroxysmal anxiety [146]. This pathway was also found to be important in our study, in which many bacteria isolated from the oral cavity (*e.g., Streptococcus parasanguinis*, *Streptococcus salivarius*) showed strong positive associations with peripheral tryptophan levels (Table 2) [147]. Tryptophan/kynurenine metabolism mediates glutamatergic neurotransmission (NMDA receptor-dependent, N-methyl-D-aspartic acid) through a pro-inflammatory immune response associated with activation of the enzyme indoleamine 2.3-dioxygenase (IDO), resulting in increased production of kynurenic acid in the brain and an imbalance in glutamatergic neurotransmission, leading to NMDA antagonism in schizophrenia [123]. Proposed neurochemical models inducing behavioural effects by blocking neurotransmission at NMDA receptors based on the action of phencyclidine (PCP), ketamine or related drugs are well known in rodents. On the other hand, the endogenous NMDA receptor antagonist kinurenic acid (KYNA), which is a metabolic product of the tryptophan-kynurenine (KP) pathway, affects glutamate release in a similar manner to the above-mentioned psychotomimetic agents, such as PCP, thus substantiating the molecular basis for its involvement in the pathophysiology of schizophrenia [148]. The tryptophan–kynurenine pathway is thought to be an important target of the mechanism through which the gastrointestinal microbiota can influence central nervous system function [149].

The positive correlation we obtained between *L. acidophilus* and blood levels of serotonin or tryptophan [Table 2] supports the notion that tryptophan metabolites formed by proteolysis by these bacteria are some of the key components of host health. Indeed, it has been suggested that these metabolites can activate the immune system, strengthen the intestinal barrier and stimulate gastrointestinal motility, as well as influence the secretion of intestinal hormones, and are, therefore, attributed with an anti-inflammatory role in the systemic circulation and modulating the composition of the intestinal microbiota. Tryptophan catabolites influence various physiological processes, promoting intestinal and systemic balance in health and disease [150]. The associations obtained in the present study between selected components of the oral microbiota and blood tryptophan levels (Table 2) indicate that the gut microflora may influence systemic inflammatory responses through metabolites involving host receptors [151].

Our study also highlighted the strong associations of individual oral microflora bacteria with individual components of the psychological assessment of the occurrence of childhood trauma. The literature describes associations of childhood trauma with selected regions of the hippocampus sensitive to the effects of disease risk stressors [152]. The association of childhood trauma with hippocampal volume reduction is found in both patients with schizophrenia and bipolar affective disorder. The magnitude of reduction is greater in patients with schizophrenia, particularly in the hippocampal outflow, pre-basal and sub-basal areas [153]. The negative correlation obtained between P.ngs and early childhood trauma further indicates that psychological treatment proposals focusing on the aftermath of childhood trauma and/or childhood abuse may affect microbial dysbiosis by restoring the natural microbiome. Stress in early life can exert long-term changes in the brain and microbiota, increasing the risk of developing depression in later life. There are few papers that show the impact of early life stress on the microbiota–gut–brain axis, but there are, nevertheless, attempts to transfer the paradigm of the impact of maternal separation stress in early life from animal models to humans [154]. Our results concerning the negative correlation of selected oral microbiome species with neurotransmitter concentrations of serotonin (5-HT [ng/mL]) and tryptophan (TRP [ug/mL]) and individual components of the childhood trauma questionnaire (CTQ EN, EA) provide support for the concept of childhood trauma-dependent depression raised above. In addition, the negative correlation we obtained between *Lactobacillus acidophilus* levels and the total score on the pre-morbid adjustment scale (PASG_B) provides support for the theory that the pre-morbid period depends on how early disruptions in the host microbiota occurred, influencing the actual timing of schizophrenia development, while determining the more complicated course of the illness in later life [155] as a key predictor of clinical and psychosocial end result [156].

A strong negative correlation with *Lactobacillus acidophilus*., L.ac, was observed for oral microbiota bacteria, which showed associations with the severity of depression, as assessed on the Hamilton scale in patients with schizophrenia (r_s_ = −0.52; *p* = 0.001). The results are consistent with those of a randomized controlled clinical trial, in which the administration of a probiotic containing, inter alia, *Lactobacillus acidophilus* (2 × 10^9^ CFU/g) to patients with major depressive disorder (MDD) for 8 weeks had a favorable effect on the Beck Depression Inventory, in addition to an effect on the levels of inflammatory and antioxidant indices, i.e., hs-CRP and glutathione [157]. In opposition are the results of Dickerson et al. (2014), which show that 14 weeks supplementation with *Lactobacillus rhamnosus* and *Bifidobacterium* has no beneficial effect in people with schizophrenia [158].

## 5. Conclusions

This study is the first to demonstrate the association of the oral microbiota with the effects of acid stress induced by an increase of brain lactate in schizophrenia patients for whom negative and/or cognitive symptoms appeared to be predominant.

The findings presented here may unveil a new interpretive framework for understanding microbial dysbiosis associated with schizophrenia. We hypothesise that the diverse oral microflora lead to differential relationships between microorganisms and their metabolites and brain metabolic activity, being the cause of disturbed brain redox balance, including altered glutamatergic neurotransmission and leading to NMDA antagonism ultimately manifesting in the negative and/or cognitive symptoms of schizophrenia.

The results of our study and the range of correlations obtained support a diverse symptomatology and the occurrence of phenotypically distinct subgroups of patients with schizophrenia. Regardless of the clinical presentation and stage of the illness, molecular biomarkers determined by neuroimaging methods may facilitate the diagnosis of neurobiologically distinct subgroups of patients with schizophrenia, which may represent an important step forward in differentiating patient subtypes in order to study the neurobiological basis of the illness and as a target for future therapies in clinical trials.

The associations obtained in the present study prompt a hypothesis indicating that changes in the oral microflora may be considered as one important diagnostic and therapeutic strategy for the diagnosis and pharmacotherapy of phenotypically distinct subgroups of patients with schizophrenia.

The present study carries several limitations, including that we cannot conclusively establish a causal relationship between the oral microbiota and NMDA receptor function in schizophrenia or the disease itself. Further, the actual microbial gene content of the isolated oral material is unknown. Dental assessment of the dental, gingival and oral hygiene status of the study participants was omitted, and the effect of diet on peripheral and central metabolites related to lactate or alanine levels in the blood and brain of the patients was not considered.

A limitation of the conducted studies using MALDI TOF-MS is the occurrence of species in the oral cavity that are difficult to culture or that require special culture conditions. Therefore, the bacterial species shown in this study as living representatives of the oral microflora may not coincide with those reported in studies using sequencing [159]. The downside of this approach is also that the use of a limited number of culture media commonly used in clinical microbiology laboratories means that some bacteria may have been missed due to selection bias. Another limitation of this approach is the relatively short incubation time of the culture media; they were cultured for a maximum of 48 h, which is too short a time for slow-growing microorganisms.

The results are encouraging to expand the study to assess the microbiota–microbiome–mouth–gut–brain pathway in patients with schizophrenia endophenotypes to determine the causal relationship between the microbiota–microbiome–brain and the clinical phenotype of the disease. Assessing the interactions between the oral–gut microbiota and their interactions with the microbiome and the brain may be a step forward in understanding these interactions. In addition, it is important to apply biostatistical methods that take into account a large number of variables—brain, laboratory and clinical—in order to discover new objective, measurable biomarkers that can facilitate the diagnosis and monitoring of the course of different forms of schizophrenia, as well as their effective treatment.

## Figures and Tables

**Figure 1 biomedicines-11-00240-f001:**
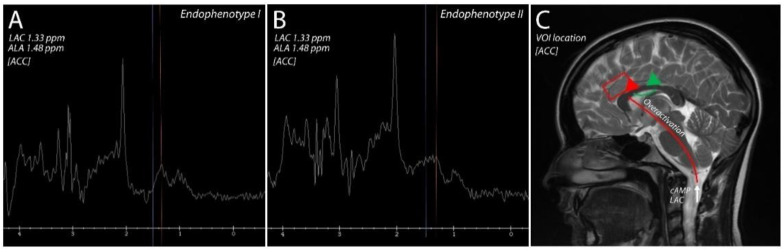
Spectroscopic spectrum. LAC–lactate-derived signal, ALA—alanine. (**A**) - Endophenotype I, (**B**) - Endophenotype II, (**C**) - VOI location.

**Figure 2 biomedicines-11-00240-f002:**
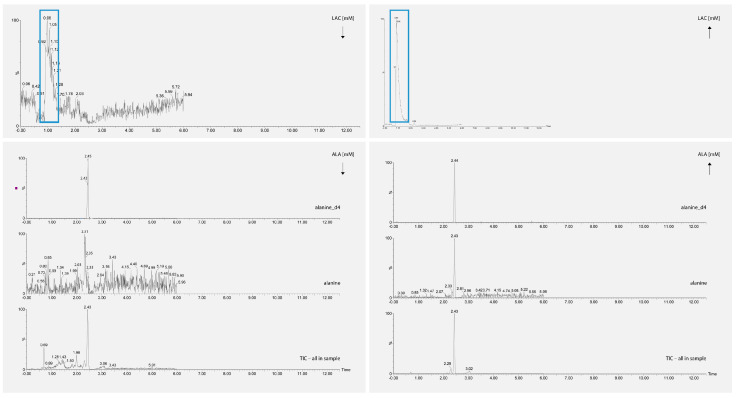
Chromatograms of selected metabolites, alanine (ALA) and lactates (LAC) in peripheral blood by LC-ESI-MS/MS.

**Figure 3 biomedicines-11-00240-f003:**
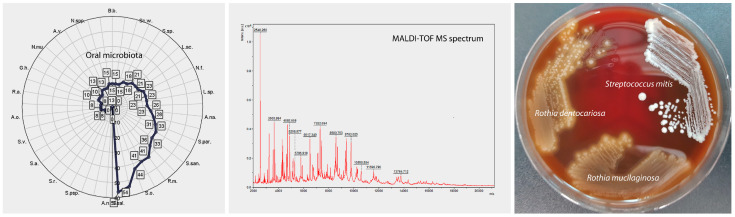
Protocol for the Isolation and Identification of Oral Microbiota by Matrix-Assisted Laser Desorption Ionization (MALDI-TOF MS).

**Figure 4 biomedicines-11-00240-f004:**
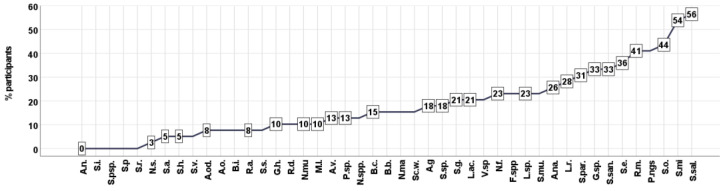
Prevalence of particular species of bacteria isolated from cheek and dorsal swabs of the tongue and saliva samples from patients with schizophrenia. A.v.—*Actinomyces viscosus*, P.sp.—*Prevotella* sp., N.spp.—*Neisseria* spp., B.c.—*Bacillus circulans*, B.b.—*Brevibacillus brevis*, N.ma.—*Neisseria macacae*, Sc.w.—*Scardovia wiggsiae,* A.g.—*Actinomyces graevenitzii,* S.sp.—*Streptococcus* sp., S.g.—*Streptococcus gordonii*, L.ac.—*Lactobacillus acidophilus*, V.sp.—*Veilonella* sp., N.f.—*Neisseria flavescens,* F.spp—*Fusobacterium* spp., L.sp.—*Leptotrichia* sp., S.mu.—*Streptococcus mutans*, A.na—Actinomyces naeslundi, L.r.—*Lactobacillus rhamnosus*, S.par.—*Streptococcus parasanguinis*, G.sp.—*Gemella* sp., S.san.—*Streptococcus sanguinis*, S.e.—*Staphylococcus epidermidis*, R.m.—*Rothia mucilaginosa*, P.ngs—*Prevotella nigrescens*, S.o.—*Streptococcus oralis*, S.mi—*Streptococcus mitis*, S.sal.—*Streptococcus salivarius*.

**Figure 5 biomedicines-11-00240-f005:**
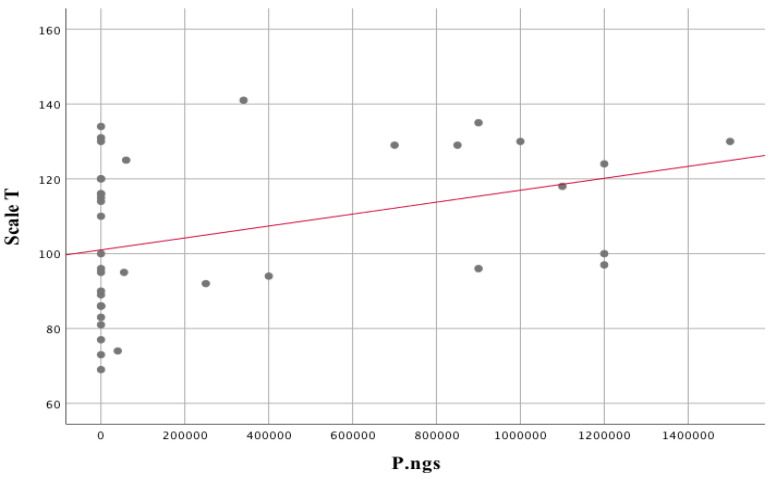
Relationship between *Prevotella nigrescens* (P.ngs) and the subjects’ Total score on the Positive and Negative Syndrome Scale (PANSS-T).

**Figure 6 biomedicines-11-00240-f006:**
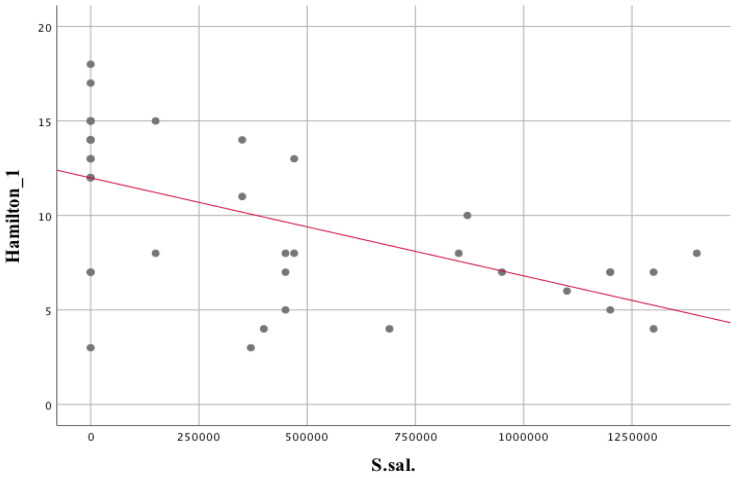
Relationship between *Streptococcus salivarius* (S.sal) and the subjects’ Hamilton scale score.

**Figure 7 biomedicines-11-00240-f007:**
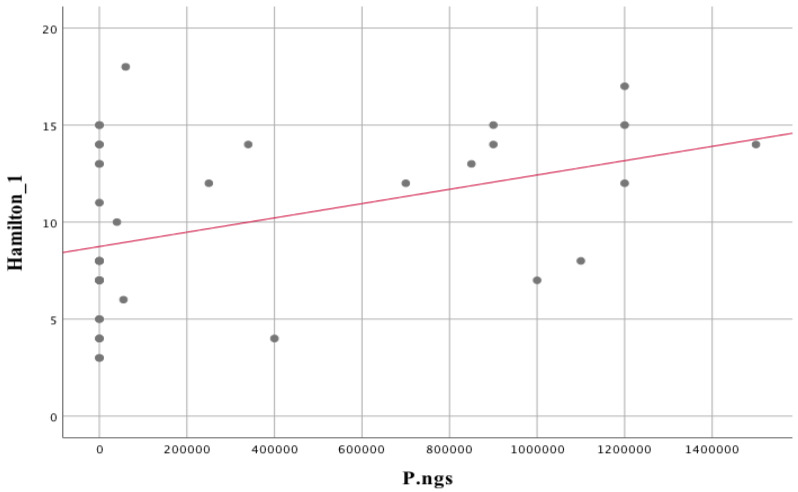
Relationship between *Prevotella nigrescens* (P.ngs) and the subjects’ score on the Hamilton scale.

**Table 1 biomedicines-11-00240-t001:** Descriptive statistics for the primary endpoint of improved total score on the Positive and Negative Syndrome Scale (PANSS-T) in the identified clusters.

Concentration	M	Me	SD	Min	Max	Q1	Q3	Statistical Test Result *
**No. 1**	124.7	126	8.24	110	141	116.5	130	U = 0; *p* < 0.001; eta^2^ = 0.74
**No. 2**	87.95	89.5	9.29	69	100	81.5	95.75	

* Mann–Whitney U test; No. 1—subjects who scored higher on the T scale (cluster 1); No. 2—subjects who scored lower on the T scale (cluster 2); M—mean, Me—Median, SD—Standard Deviation, Min—Minimum, Max—Maximum, Q1–Q3—interquartile range between the 1st and 3rd quartile, eta^2^ is a measure of effect size that was used in ANOVA models. It measured the proportion of variance associated with each main effect and interaction effect in an ANOVA model.

**Table 2 biomedicines-11-00240-t002:** Significant correlation analysis of all variables analyzed in the study.

rho Spearmana	Parameter	Correlation Coefficient	Bilateral Relevance
** *Prevotella nigrescens* **			
	Hamilton scale	r_s_ = 0.422	*p* = 0.007
	Duration of the disease	r_s_ = 0.343	*p* = 0.033
	Scale T	r_s_ = 0.372	*p* = 0.020
	CTQ EN	r_s_ = −0.392	*p* = 0.013
	CTQ EA	r_s_ = −0.390	*p* = 0.014
	CTQ M	r_s_ = −0.378	*p* = 0.018
	5-HT [ng/mL]	r_s_ = −0.374	*p* = 0.019
	TRP [ug/mL]	r_s_ = −0.509	*p* = 0.001
	ALA [mM]	r_s_ = 0.496	*p* = 0.001
***Veilonella* spp.**			
	Hamilton scale	r_s_ = 0.453	*p* = 0.004
	5-HT [ng/mL]	r_s_ = −0.386	*p* = 0.015
	LAC [mM]	r_s_ = 0.344	*p* = 0.032
	ALA [mM]	r_s_ = 0.409	*p* = 0.010
** *Actinomyces graevenitzii* **			
	*Veilonella* spp.	r_s_ = 0.371	*p* = 0.020
	*Strepptococcus gordonii*	r_s_ = 0.438	*p* = 0.005
	*Staphylococcus epidermidis*	r_s_ = 0.331	*p* = 0.039
	*Rothia mucilaginosa*	r_s_ = 0.419	*p* = 0.009
	*Prevotella* spp.	r_s_ = 0.511	*p* = 0.001
	*Neisseria macacae*	r_s_ = 0.509	*p* = 0.001
	*Lactobacillus rhamnosus*	r_s_ = 0.385	*p* = 0.016
	*Leptotrichia* sp.	r_s_ = 0.327	*p* = 0.042
	*Gemella* spp.	r_s_ = 0.393	*p* = 0.013
	*Actinomyces odontolyticus*	r_s_ = 0.414	*p* = 0.009
	*Actinomyces naeslundii*	r_s_ = 0.777	*p* = 0.001
	ACC_LAC	r_s_ = 0.360	*p* = 0.024
***Neisseria* spp.**			
	CTQ SA	r_s_ = 0.327	*p* = 0.042
	PAS_LA	r_s_ = −0.437	*p* = 0.014
	5-HT [ng/mL]	r_s_ = 0.413	*p* = 0.009
	TRP [ug/mL]	r_s_ = 0.487	*p* = 0.002
	ALA [mM]	r_s_ = −0.350	*p* = 0.029
***Fusobacterium* spp.**			
	New_N	r_s_ = −0.337	*p* = 0.036
	neg_1	r_s_ = −0.328	*p* = 0.041
** *Lactobacillus rhamnosus* **			
	Hamilton scale	r_s_ = −0.466	*p* = 0.003
	STAI_Cecha1	r_s_ = −0.324	*p* = 0.044
	5-HT [ng/mL]	r_s_ = −0.713	*p* = 0.001
	TRP [ug/mL]	r_s_ = −0.486	*p* = 0.002
	ALA [mM]	r_s_ = −0.392	*p* = 0.013
	*Actinomyces graevenitzii*	r_s_ = 0.385	*p* = 0.016
	*Actinomyces viscosus*	r_s_ = 0.323	*p* = 0.045
	*Prevotella* sp.	r_s_ = 0.679	*p* = 0.001
	*Streptococcus parasanguinis*	r_s_ = 0.404	*p* = 0.011
	*Streptococcus salivarius*	r_s_ = 0.389	*p* = 0.014
** *Lactobacillus acidophilus* **			
	Hamilton scale	r_s_ = −0.617	*p* = 0.001
	Number of episodes	r_s_ = 0.356	*p* = 0.026
	MoCa	r_s_ = −0.394	*p* = 0.013
	PASGB	r_s_ = −0.429	*p* = 0.016
	5-HT [ng/mL]	r_s_ = 0.428	*p* = 0.007
	TRP [ug/mL]	r_s_ = 0.394	*p* = 0.013
	ALA [mM]	r_s_ = −0.614	*p* = 0.001
	*Bacillus circulans*	r_s_ = 0.534	*p* = 0.001
	*Staphylococcus epidermidis*	r_s_ = −0.365	*p* = 0.022
	*Streptococcus parasanguinis*	r_s_ = 0.506	*p* = 0.001
** *Gemella haemolysans* **			
	DUP	r_s_ = 0.335	*p* = 0.037
	T scale	r_s_ = 0.339	*p* = 0.035
	exc_1	r_s_ = 0.328	*p* = 0.041
	STAIcecha_1	r_s_ = −0.365	*p* = 0.022
	STAIstan_1	r_s_ = 0.348	*p* = 0.030
	STAIcecha_1	r_s_ = 0.365	*p* = 0.022
	CTQ_SA	r_s_ = 0.438	*p* = 0.005
	*Streptococcus vestibularis*	r_s_ = 0.333	*p* = 0.038
***Leptotrichia* sp. **			
	Hamilton scale	r_s_ = −0.466	*p* = 0.003
	T scale	r_s_ = 0.317	*p* = 0.049
	Calgary	r_s_ = −0.332	*p* = 0.039
	LAC [mM]	r_s_ = 0.665	*p* < 0.001
	ALA [mM]	r_s_ = 0.340	*p* = 0.034
	ACC Glutamate	r_s_ = 0.406	*p* = 0.013
	*Actinomyces graevenitzii*	r_s_ = 0.327	*p* = 0.042
	*Actinomyces naeslundi*	r_s_ = 0.654	*p* < 0.001
	*Gemella* sp.	r_s_ = 0.658	*p* < 0.001
	*Neisseria flavescens*	r_s_ = 0.466	*p* < 0.001
	*Neisseria macacae*	r_s_ = 0.603	*p* < 0.001
	*Rothia dentocariosa*	r_s_ = 0.416	*p* = 0.008
	*Rothia mucilaginosa*	r_s_ = 0.554	*p* < 0.001
	*Scardovia wiggsiae*	r_s_ = 0.591	*p* < 0.001
	*Staphylococcus epidermidis*	r_s_ = 0.492	*p* = 0.001
	*Staphylococcus haemolyticus*	r_s_ = 0.434	*p* = 0.006
	*Streptococcus mutans*	r_s_ = 0.380	*p* = 0.017
	*Streptococcus oralis*	r_s_ = 0.403	*p* = 0.011
** *Streptococcus sobrinus* **			
	Scale T	r_s_ = 0.339	*p* = 0.035
	CTQ_M	r_s_ = −0.334	*p* = 0.038
	*Streptococcus* sp.	r_s_ = 0.404	*p* = 0.011
	*Prevotella nigrescens*	r_s_ = 0.423	*p* = 0.007
** *Streptococcus salivarius* **			
	Hamilton scale	r_s_ = −0.594	*p* = 0.000
	exc_1	r_s_ = −0.399	*p* = 0.012
	PASG_B	r_s_ = 0.401	*p* = 0.026
	5-HT [ng/mL]	r_s_ = 0.506	*p* = 0.001
	TRP [ug/mL]	r_s_ = 0.353	*p* = 0.027
	ALA [mM]	r_s_ = −0.541	*p* = 0.000
	*Leptotrichia* sp.	r_s_ = −0.365	*p* = 0.022
	*Lactobacillus rhamnosus*	r_s_ = 0.389	*p* = 0.014
	*Rothia mucilaginosa*	r_s_ = −0.408	*p* = 0.011
	*Staphylococcus epidermidis*	r_s_ = −0.361	*p* = 0.024
	*Streptococcus mutans*	r_s_ = −0.339	*p* = 0.035
	*Streptococcus oralis*	r_s_ = −0.371	*p* = 0.020
	*Streptococcus parasanguinis*	r_s_ = 0.456	*p* = 0.004
	*Streptococcus* sp.	r_s_ = 0.321	*p* = 0.046
** *Streptococcus parasanguinis* **			
	Hamilton scale	r_s_ = −0.746	*p* = 0.000
	5-HT [ng/mL]	r_s_ = 0.450	*p* = 0.003
	TRP [ug/mL]	r_s_ = 0.404	*p* = 0.011
	LAC [mM]	r_s_ = −0.334	*p* = 0.038
	ALA [mM]	r_s_ = −0.610	*p* = 0.000
	*Bacillus circulans*	r_s_ = 0.432	*p* = 0.006
	*Leptotrichia* sp.	r_s_ = −0.353	*p* = 0.027
	*Lactobacillus rhamnosus*	r_s_ = 0.404	*p* = 0.011
	*Lactobacillus acidophilus*	r_s_ = 0.506	*p* = 0.001
	*Staphylococcus epidermidis*	r_s_ = −0.403	*p* = 0.011
	*Streptococcus salivarius*	r_s_ = 0.456	*p* = 0.004
** *Streptococcus mitis* **			
	CTQ_SA	r_s_ = −0.347	*p* = 0.031
	PAST_B	r_s_ = −0.390	*p* = 0.030
	GLUT [µg/mL]	r_s_ = 0.320	*p* = 0.047
	*Streptococcus mutans*	r_s_ = 0.352	*p* = 0.028
	*Streptococcus sanguinis*	r_s_ = 0.465	*p* = 0.003
** *Rothia mucilaginosa* **			
	Hamilton scale	r_s_ = 0.446	*p* = 0.005
	PAS_A	r_s_ = −0.362	*p* = 0.049
	GLUT [µg/mL]	r_s_ = 0.337	*p* = 0.038
	ALA [mM]	r_s_ = 0.452	*p* = 0.004
	*Actinomyces graevenitzii*	r_s_ = 0.419	*p* = 0.009
	*Actinomyces naeslundi*	r_s_ = 0.489	*p* = 0.002
	*Gemella* sp.	r_s_ = 0.418	*p* = 0.009
	*Leptotrichia* sp.	r_s_ = 0.554	*p* = 0.000
	*Neisseria macacae*	r_s_ = 0.486	*p* = 0.002
	*Rothia dentocariosa*	r_s_ = 0.341	*p* = 0.036
	*Scardovia wiggsiae*	r_s_ = 0.502	*p* = 0.001
	*Staphylococcus epidermidis*	r_s_ = 0.608	*p* = 0.000
	*Streptococcus mutans*	r_s_ = 0.343	*p* = 0.035

**Table 3 biomedicines-11-00240-t003:** Descriptive statistics for S.sal. S.par. and S.e. in the study group by severity of depressive disorder. Eight subjects (20.5%) had no depression, 17 had mild depression (43.6%) and the remaining 14 (35.9%) had moderate depression. Oral bacterial species confirmed by the MALDI-TOF method.

Variable		M	Sd	Min	Max	Q1	Me	Q3
S.sal.	No depression	688,750	466,060	0	1,300,000	377,500	570,000	1,175,000
	Mild depression	567,058	513,344	0	1,400,000	0	450,000	1,075,000
	Moderate depression	69,286	151,579	0	470,000	0	0	37,500
S.par.	No depression	127,500	55,485	30,000	180,000	72,500	150,000	160,000
	Mild depression	28,823	52,715	0	170,000	0	0	30,000
	Moderate depression	0	0	0	0	0	0	0
S.e.	No depression	0	0	0	0	0	0	0
	Mild depression	219,412	426,856	0	1,200,000	0	0	270,000
	Moderate depression	634,286	592,618	0	1,800,000	0	775,000	1,125,000

## Data Availability

The data are available from the corresponding authors upon reasonable request.

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
