# Peer review of "The Association of the Oral Microbiota with the Effects of Acid Stress Induced by an Increase of Brain Lactate in Schizophrenia Patients"

_biomedicines, 2023, doi:10.3390/biomedicines11020240_

Round 1
Reviewer 1 Report
Congratulations to the authors for this manuscript, which is of great interest although it is too complex and needs major revision to be accepted in Biomedicines.
The authors present a large number of results, with tables appearing on 4 consecutive pages of the document. The capacity for synthesis is important, carrying out an exhaustive analysis to have a stronger message with only the necessary data to understand the manuscript, and the rest of the results should be moved to supplementary material.
The same happens with the number of citations, which is high for an article and could perhaps be reduced, but in this case is not limited the number of cites.
This can be applied to other points in the manuscript. For example, the journal indicates in its recommendations for authors "Abstract: The abstract should be a total of about 200 words maximum." and it has 363 words.
For all this, the authors have to redo the manuscript to make the manuscript easier to read, reducing everything they do not consider essential, moving results to supplementary material in order to re-evaluate a manuscript that is extremely complex to interpret, and therefore requires a major revision.
Author Response
Dear Reviewer,
Please find attached a corrected version of our paper " The association of the oral microbiota with the effects of acid stress induced by an increase of brain lactate in schizophrenia patients “, an original paper by KrzyÅ›ciak W. et al. We would like to thank the Reviewer for their generally positive assessment of our manuscript and for their helpful comments. We have tried to follow all of them and we have been able to fulfill almost all the Reviewers’ requirements. We believe this has allowed us to improve the quality of the paper.
You can find our responses to the Reviewers' comments on the website:
https://drive.google.com/file/d/1GdrMTyyDqcQi2y7UDp7muE-UqnRXwU2H/view?usp=share_link
We hope that in the current version the manuscript is clearer and meets the Editorial requirements.
If You have any further questions, We are open to discussion.
Thank You for Your time and comments.
Yours faithfully,
Authors

Reviewer 2 Report
Dear authors
The article titled “The association of the oral microbiota with the effects of acid stress induced by an increase of brain lactate in schizophrenia patients” is a well-written and very interesting article that highlights the potential association of acid stress in the brain and gastrointestinal microbiota in schizophrenia. The data is well analyzed however, the manuscript should go under general refinement according to some comments below
1. In material and methods “rpm” should be reported in “g”
2. Authors mentioned that serum samples were analyzed by LC-ESI-MS/MS using a method by Zhe et al., (2017) with in-house modifications [71].
Comment: instead of “2017” there should be a reference number instead of the year. A brief description of in-house modification would be good to mention here.
3. What was the rationale behind using MALDI-TOF for the identification of oral microorganisms instead of using 16s RNA sequencing and how reliable is MALDI-TOF as compared to 16s RAN sequencing? Add it in the section “Isolation of oral microorganisms and identification by matrix-assisted laser desorption ionization
(MALDI-TOF MS)”.
4. In table 1 legend, the authors have mentioned the description of abbreviations like Me-Median….. Add a description of “SD” as well.
5. There are so many full stops in section 3.4 of results, in the discussion and conclusion part in the middle of sentences. Remove all unnecessary full stops and make proper sentences. A few full stops and some comments have been highlighted in the manuscript attached
6. Reference 97 “carboxylase”

Author Response
December 28th, 2022
Letter to the Reviewer
Dear Reviewer,
Please find attached a corrected version of our paper " The association of the oral microbiota with the effects of acid stress induced by an increase of brain lactate in schizophrenia patients “, an original paper by KrzyÅ›ciak W. et al. We would like to thank the Reviewer for their generally positive assessment of our manuscript and for their helpful comments. We have tried to follow all of them and we have been able to fulfill almost all the Reviewers’ requirements.
You can find our responses to Your comments on the website:
https://drive.google.com/file/d/1dTa3akKhlPRX7fobgg8eR0XvQTkT9TE0/view?usp=share_link
We hope that in the current version the manuscript is clearer and meets the Editorial requirements.
If You have any further questions, We are open to discussion.
Thank You for Your time and comments.
Yours faithfully,
Authors

Reviewer 3 Report
-correction of the work by a person specialized in English -checking the way of writing the bibliography (eg: 158)Author Response
December 28th, 2022
Letter to the Reviewer
Dear Reviewer,
Please find attached a corrected version of our paper " The association of the oral microbiota with the effects of acid stress induced by an increase of brain lactate in schizophrenia patients “, an original paper by KrzyÅ›ciak W. et al. We would like to thank the Reviewer for their generally positive assessment of our manuscript and for their helpful comments. We have tried to follow all of them and we have been able to fulfill almost all the Reviewers’ requirements. We believe this has allowed us to improve the quality of the paper.
We would be grateful if You consider our revised manuscript for publication in Biomedicines. In any case, we would like to let You know that we are open to consideration of any further comments on our answers.
Dear Reviewer,
thank You for the positive evaluation of our manuscript and the valuable comments that we have tried to include in the manuscript. We hope that the paper meets the expectations of the Reviewer and the journal's requirements in its current form. We are open to further discussion and suggestions for the current form of the manuscript.
The paper has been re-checked by an English Native Speaker. The way the bibliography is written has been checked according to the journal's requirements throughout the manuscript. We hope that, in its current form, the manuscript meets the expectations of the Reviewer and, with suggested changes to the reported comments of the three Reviewers, is also suitable for publication in the journal Biomedicines.
We hope that in the current version the manuscript is clearer and meets the Editorial requirements.
If You have any further questions, We are open to discussion.
Thank You for Your time and comments.
Yours faithfully,
Authors

Round 2
Reviewer 1 Report
Congratulations to the authors for this new version of the manuscript, which must be accepted in the journal Biomedicines.
The new version has improved a lot, making it much easier to read. From my point of view, perhaps some point such as the conclusions could be reduced, reaching a higher level of specificity. For example, the first conclusion stated by the authors is "This study is the first to demonstrate the association of the oral microbiota with the effects of acid stress induced by an increase of brain lactate in schizophrenia patients for whom negative and/or cognitive symptoms appeared to be predominant" or "A limitation of the conducted studies using MALDI TOF-MS ..." For me, the first point is a characteristic of the study rather than a conclusion, and the second is a characteristic of the technique used that perhaps should not appear in the conclusions point, but it may be questionable whether or not it is conclusive to be included in this section. In any case, this can no longer be considered defects for not accepting the manuscript, but rather personal appreciations about publication styles. The abstract has also greatly improved, adjusting to the indications of Biomedicines, as the reduction of tables and not relevant information.
For all this, I ratify my congratulations to the authors for this new version, which must be accepted in Biomedicines.